# Understanding microRNA-Mediated Chemoresistance in Colorectal Cancer Treatment

**DOI:** 10.3390/ijms26031168

**Published:** 2025-01-29

**Authors:** Guillermo Valenzuela, Héctor R. Contreras, Katherine Marcelain, Mauricio Burotto, Jaime González-Montero

**Affiliations:** 1Basic and Clinical Oncology Department, Faculty of Medicine, University of Chile, Santiago 8350499, Chile; guillermo.valenzuela@ug.uchile.cl (G.V.); hcontrer@uchile.cl (H.R.C.); kmarcelain@uchile.cl (K.M.); 2Center for Cancer Prevention and Control (CECAN), Santiago 8380453, Chile; 3Bradford Hill Clinical Research Center, Santiago 8380453, Chile; mburotto@bh.cl

**Keywords:** colorectal cancer, microRNA, drug resistance, fluorouracil, oxaliplatin

## Abstract

Colorectal cancer (CRC) remains the second most lethal cancer worldwide, with incidence rates expected to rise substantially by 2040. Although biomarker-driven therapies have improved treatment, responses to standard chemotherapeutics, such as 5-fluorouracil (5-FU), oxaliplatin, and irinotecan, vary considerably. This clinical heterogeneity emphasizes the urgent need for novel biomarkers that can guide therapeutic decisions and overcome chemoresistance. microRNAs (miRNAs) have emerged as key post-transcriptional regulators that critically influence chemotherapy responses. miRNAs orchestrate post-transcriptional gene regulation and modulate diverse pathways linked to chemoresistance. They influence drug transport by regulating ABC transporters and affect metabolic enzymes like thymidylate synthase (TYMS). These activities shape responses to standard CRC chemotherapy agents. Furthermore, miRNAs can regulate the epithelial–mesenchymal transition (EMT). The miR-200 family (e.g., miR-200c and miR-141) can reverse EMT phenotypes, restoring chemosensitivity. Additionally, miRNAs like miR-19a and miR-625-3p show predictive value for chemotherapy outcomes. Despite these promising findings, the clinical translation of miRNA-based biomarkers faces challenges, including methodological inconsistencies and the dynamic nature of miRNA expression, influenced by the tumor microenvironment. This review highlights the critical role of miRNAs in elucidating chemoresistance mechanisms and their promise as biomarkers and therapeutic targets in CRC, paving the way for a new era of precision oncology.

## 1. Introduction

Colorectal cancer (CRC) is the malignancy with the second highest cancer mortality rate in the world. Its global incidence is projected to continue to increase progressively up to 2040 [1,2]. Despite advances in early diagnosis, some patients with advanced cancer still have a poor prognosis. Patients with advanced or stage IV cancer are characterized by advanced disease with metastasis to other organs, and have a poor life expectancy [3]. In this context, personalized markers have been sought for the selection of optimal therapies in these patients with advanced cancer. Currently, international clinical guidelines only recommend the study of mutations in KRAS (Kirsten Rat Sarcoma Viral Oncogene Homolog), NRAS (Neuroblastoma Rat Sarcoma Viral Oncogene Homolog) and BRAF (Neuroblastoma Rat Sarcoma Viral Oncogene Homolog), along with the study of DNA mismatch repair proteins, to determine the presence or absence of microsatellite instability (MSI) [4,5]. In the case of patients with wild-type (wt) KRAS, treatment with endothelial growth factor receptor (EGFR) inhibitors, such as cetuximab or panitumumab, is an option [6,7], while patients with high microsatellite instability (MSI-H) are candidates for immunotherapy [8]. Despite these advances, only approximately 40% and 13% of patients have KRAS wt and MSI-H phenotypes [9], respectively. Thus, 47% of patients with stage IV CRC are not candidates for these therapies and undergo conventional chemotherapy with regimens containing 5-fluorouracil (5-FU) or capecitabine with oxaliplatin and/or irinotecan (FOLFOX/XELOX or FOLFIRI regimen). Within this group, clinical guidelines recommend considering the laterality of the colon, especially the right colon, as a marker for prescribing intensified triplet chemotherapy (FOLFOXIRI) + bevacizumab. However, the right colon is the least common location for cancer, and few patients are in a clinical condition suitable to withstand the intensity of this regimen [10]. Patients undergoing conventional chemotherapy have a variable response rate, with some showing long-lasting responses and others experiencing rapid tumor progression and mortality [11]. One approach to addressing this problem is searching for biomarkers of prognosis (e.g., MSI) and response to treatment (KRAS mutations for anti-EGFR use). Although such biomarkers exist, their implementation in clinical practice is deficient. Although many potential biomarkers are described in the scientific literature, they do not fit into the clinical context, and the experimental design of the relevant studies makes their routine clinical use to guide conventional chemotherapies unfeasible [12,13].

Identifying biomarkers has benefits for patients. For example, they help in the prescription of more effective chemotherapy combinations and prevent exposure to adverse drug effects [14]. Apart from benefitting patients, biomarkers can allow health systems to use resources optimally; for example, KRAS testing prior to anti-EGFR use is shown to be more cost-effective than treating patients universally with these drugs [15]. Therefore, it is necessary to identify biomarkers that allow for predicting the response to the best chemotherapeutic combinations. Thus, following bioinformatics studies of transcriptome sequencing, research on non-coding RNA sequences, such as microRNA (miRNA), long non-coding RNA (lncRNA), and circRNA, as preventive markers has gained interest.

## 2. Metabolic Aspects of Chemotherapy in Colorectal Cancer

The first-line regimen in mCRC is the combination of 5-FU + leucovorin (LV) with oxaliplatin (FOLFOX), capecitabine + oxaliplatin (XELOX), or irinotecan (FOLFIRI).

### 2.1. 5-Fluorouracil/Leucovorin

5-FU exerts its cytotoxic effects by metabolizing its phosphorylated metabolites, preventing adequate DNA and RNA synthesis [16]. The drug enters through passive-diffusion transporters, mainly organic cation transporter 2 (OAT2), also known as SLC22A7 [17]. In the cell, 5-FU is metabolized by a series of enzymes. One of the steps, converting 5-FU to fluorodeoxyuridine (FdUR), is catalyzed by the enzyme thymidylate synthase (TYMS). Subsequently, FdUR is phosphorylated, forming fluorodeoxyuridine monophosphate (FdUMP). This metabolite irreversibly inhibits the action of TYMS by forming stable compounds with 5,10-methylenetetrahydrofolate [18]. As a consequence, a nucleotide imbalance is generated, where TYMS is not able to metabolize deoxyuridine monophosphate (dUMP) to deoxythymidine monophosphate (dTMP), resulting in cellular arrest and inability to repair and synthesize DNA [19]. FdUMP is also phosphorylated to FdUTP, which cannot be incorporated into DNA. Alternatively, RNA synthesis is prevented by successive phosphorylations of 5-FU to fluorouridine monophosphate (FUMP), which, being fluorinated compounds, will not be able to be incorporated into RNA [19].

LV (folinic acid) is a 5-formyl derivative of folic acid, which plays a role in enhancing the effect of 5-FU. LV (5-formyltetrahydrofolate) increases the levels of 5,10-methylenetetrahydrofolate at the intracellular level. This compound binds to TYMS, allowing its activity to be irreversibly inhibited [20]. This theoretical molecular effect has been shown to improve clinical outcomes: a meta-analysis comparing 5-FU with 5-FU + leucovorin treatment found objective response rates of 11% and 23%, respectively [21].

### 2.2. Capecitabine

Capecitabine belongs to the class of fluoropyrimidines and is a prodrug that is metabolized into 5-FU. It has the advantage of an oral administration route, bypassing the need for central venous catheterization. This is because capecitabine is not metabolized in the gastrointestinal tract and is absorbed intact into the bloodstream [22]. The conversion of capecitabine to 5-FU occurs in a three-step cascade, first involving carboxylesterase enzymes in the liver, following which it is hydrolyzed by cytidine deaminase in the liver or tumor. Finally, the metabolites are transformed into 5-FU by thymidine phosphorylase, which is mainly located in the tumor [23]. Clinically, the effectiveness of the XELOX regimen (capecitabine + oxaliplatin) is equivalent in mortality and objective response rate to that of the FOLFOX regimen, according to the meta-analysis of eight clinical trials [24]. Therefore, clinical guidelines state that 5FU/LV-based regimens and capecitabine + oxaliplatin are equivalent [4,5].

### 2.3. Oxaliplatin

Oxaliplatin is part of the chemotherapeutic group called the platinums, which, unlike carboplatin and cisplatin, are active in CRC [25]. Its main mechanism of action is DNA damage through the formation of intra-strand adducts between two adjacent guanine residues or between guanine and adenine, thereby preventing DNA replication and transcription [26]. Traditionally, oxaliplatin was believed to passively enter cells; however, recent evidence has described binding to copper transporters (hCTR1) and, to a lesser extent, transport through solute carrier transporters (SCLs), although in vivo evidence is less categorical about its role [27]. In DNA, the formation of adducts with different platinums differs three-dimensionally. This has an impact since translesion DNA polymerases ensure DNA replication in the absence of repair [28]. DNA polymerases beta (POLβ) and eta (POLη) are described to be more efficient in bypassing the adducts generated by cisplatin than by oxaliplatin [29]. In addition, the expression of these DNA polymerases (POLβ, POLη, and POLζ) is inversely associated with the cytotoxic effect [30]. On the other hand, nucleotide excision repair (NER) is the primary DNA repair pathway due to platinum cytotoxicity. The cell’s failure to repair will ultimately lead to cell death via apoptosis, regulated necrosis, and autophagy [27].

### 2.4. Irinotecan

Irinotecan is a compound that inhibits topoisomerase I. It forms a complex with topoisomerase and DNA, generating a bond that leads to signaling checkpoint damage, replication fork arrest, and cell death [31]. Irinotecan is a prodrug that is metabolized in the liver by the carboxylesterase enzymes (CES1 and CES2) and in the plasma by butyrylcholinesterase (hBChE), forming the active metabolite SN-38, which has a cytotoxic effect [32].

## 3. Chemotherapy Resistance Mechanisms in Colorectal Cancer

One of the variables that determines the prognosis of patients with CRC who receive chemotherapy is the development of resistance. Currently, several mechanisms associated with resistance and early relapse have been described in these patients.

### 3.1. 5-FU Resistance

Several resistance mechanisms for 5-FU have been described, the most-studied being those related to drug uptake and efflux, alterations in drug metabolism pathways, and the activation of anti-apoptotic pathways.

a.The overexpression of efflux pumps of the ABC-binding cassette family that allow these drugs to be eliminated from the intracellular environment is one of the main mechanisms of resistance to 5-FU [33]. Furthermore, although passive diffusion is one method of drug entry, it is aided by nucleoside exchange proteins. In patients with CRC, a higher expression of human equilibrative nucleoside transporter 1 (hENT1) is associated with a worse prognosis [34], probably due to the greater uptake of other nucleosides that allow cell proliferation [19].b.Clinical and basic studies have shown that a low expression of TYMS is associated with a better prognosis with 5-FU-based therapies [19]. Genetic polymorphisms in the 5′ ends of the untranslated region (5′-UTR) region of the TYMS gene, associated with double (2R) or triple (3R) repeats of 28 base pairs in tandem and the single-nucleotide polymorphism (SNP) G > C in the tandem base pair region, have been associated with chemoresistance in patients with CRC [35], particularly the allelic combinations 5′-UTR 2R/3G, 3C/3G, and 3G/3G [36]. Likewise, polymorphism has been described in the 3′-UTR region, specifically the deletion of 6 bp at position 1494 of the TYMS mRNA (rs151264360), which has been associated with lower TYMS expression [37]. Recently, an association of the rs151264360 del/del phenotype was found in the Chilean population, which has been correlated with poor survival in metastatic CRC [38]. In patients treated with 5-FU, chemoresistance is described to be generated secondary to selective pressure, where tumor clones are selected when TYMS has greater activity [39]. In summary, TYMS is a target of possible chemoresistance mechanisms, and its diverse activity could have prognostic clinical implications.c.Evasion of apoptosis is a mechanism of 5-FU resistance mediated through the activation of NF-κB and subsequent activation of STAT3 that allows the overexpression of some antiapoptotic factors, such as Bcl-2 and inhibitor of apoptosis protein (IAP) surviving [40,41], as well as the expression of anti-proliferative proteins such as cyclin D1, vascular endothelial growth factor (VEGF), and c-myc [19].

### 3.2. Oxaliplatin Resistance

The most-studied mechanism of oxaliplatin resistance involves the transcriptional factor FOXC2, part of the forkhead box family, which plays a role in promoting the epithelial–mesenchymal transition (EMT) through the MAPK/ERK pathway [42]. In cells with the EMT phenotype, greater resistance to chemotherapy due to a greater expression of efflux pumps and lower proliferative activity has been described [43]. The latter produces resistance since chemotherapy acts preferentially on cells with a higher proliferation rate [44]. In patients with CRC, the overexpression of ERCC1 proteins, which belong to the NER group, is associated with greater resistance to oxaliplatin. This is due to the ability of ERCC1 to repair DNA damaged by the chemotherapeutic agent [44]. A meta-analysis of 17 studies of patients with CRC and gastric cancer found an association of poor response in terms of PFS and OS to an oxaliplatin-based regimen for the ERCC1 rs11615C>T polymorphism in the T allele in Asians, as well as an association regarding PFS and OS in Caucasians for the rs13181T>G polymorphism in the G allele [45].

### 3.3. Irinotecan Resistance

Irinotecan acts through the inhibition of topoisomerases, which, under normal conditions, facilitate DNA unwinding to allow replication [46]. Resistance to irinotecan is not well characterized, with contradictory findings; however, the mechanisms are suggested to consist of alterations in drug metabolism, activation of the NF-κB pathway, and alterations in the structure or expression of topoisomerase I [47,48]. The ABCB1 and ABCG2 efflux pumps are also suggested to have a role in resistance, but studies with large patient samples have not been conclusive in finding an association between these proteins and prognosis [49,50].

## 4. microRNAs

miRNAs are short sequences of an average of 22 nucleotides found in non-coding regions of the genome. They allow the regulation of gene expression by binding to the 3′-UTR of the target messenger RNA (mRNA) [51]. MicroRNAs have some technical advantages, such as their detectability in multiple tissues (fresh tissue or formalin-fixed and paraffin-embedded [FFPE] tissue), blood, and ascites, among others. In addition, they act on multiple potential prognostic and therapeutic targets at the same time [52]. The biogenesis of miRNAs begins with the strand transcribed by RNA polymerase, called pri-miRNA, which acquires a three-dimensional hairpin-shaped structure in the nucleus. Subsequently, it is processed and transported by the ribonuclease Drosha (formerly RNASEN) and the exportin 5 complex, generating a pre-miRNA that reaches the cytoplasm. Here, it is processed by the Dicer ribonuclease, generating small miRNA duplexes. This mature miRNA binds to the RNA-induced silencing complex (RISC) that binds to an mRNA, stabilizing and generally inhibiting transcription [53,54]. The two miRNAs of the duplexes generated by Dicer enzyme cleavage are generally called 5p or 3p, depending on whether the pre-miRNA is cleaved in the 5′ or 3′ direction. Traditionally, one strand of the duplex is considered functional and the other transient because it only undergoes degradation [55]; however, more recent studies have shown that both the 3p and 5p strands can be functional and even have different target mRNAs [56].

## 5. microRNAs Genetic and Molecular Features of Chemotherapy Resistance in Colorectal Cancer

The study of miRNA has paved the way for the search for new prognostic markers and chemotherapy-response markers, and the feasibility of therapies based on silencing or increasing miRNA expression has been raised [57]. Some miRNAs may play a role in chemotherapy resistance, given their role in post-translationally regulating genes associated with resistance, such as chemotherapeutic influx and/or efflux pumps, apoptosis-associated proteins, and cell cycle regulators [19,58].

### 5.1. 5-Fluorouracil-Associated Resistance

In colon cancer cell cultures, miR-519c [59] and miR-142-3p [60] have been correlated with the expression of the ABCG2 transporter, which is partly responsible for 5-FU resistance. Moreover, miR-361 has a chemosensitivity effect on 5-FU through inhibition of the transcription factor FOXM1, a positive regulator of the ABCC10 and ABCC5 efflux pumps [61]. Higher expression of miR-330 in tumor tissue has been associated with greater chemosensitivity, with TYMS being one of the targets. There is an inverse relationship between higher levels of miR-330 and TYMS inhibition, which generates a greater response to 5-FU [62]. Similarly, miR-375-3p negatively regulates TYMS, which is associated with greater chemosensitivity [63]. miR-27a can regulate resistance to 5-FU, although in vitro overexpression of this molecule negatively regulates the enzyme DPYD, which metabolizes 5-FU to its inactive metabolite 5-dihydrofluorouracil [64]. Database studies (TCGA) show that miR-27a overexpression is associated with worse disease-free survival, which could be because other targets include the base excision repair proteins ERCC1 and ERCC4, associated with resistance to oxaliplatin [65]. Downregulation of miR-206 has been found in 5-FU-resistant cells, which, as a consequence, would show greater activity of the anti-apoptotic factor Bcl-2 [66].

### 5.2. Oxaliplatin-Associated Resistance

miR-143 overexpression could be associated with chemosensitivity to oxaliplatin via the inhibition of IGF-IR, a regulator of cell proliferation and survival [67]. The transcription factor FOXQ1 has been associated with resistance mechanisms through the activation of the TGF-β1 and Wnt pathways [68]. In vitro, miR-106a overexpression could increase sensitivity to oxaliplatin by inhibiting FOXQ1 [69]. In oxaliplatin-resistant cells, miR-454-3p upregulation was found to activate the PI3K/Akt pathway by inhibiting PTEN [70]. The activation of EMT pathways has also been seen as a mechanism of chemoresistance. For example, the knockdown of miR-23b was described to restore chemosensitivity in oxaliplatin-resistant cell lines by decreasing EMT markers such as SNAI2 and vimentin [71]. Moreover, members of the miR-200 family (miR-200c and miR-141) are downregulated in oxaliplatin-resistant cell lines and are associated with the expression of EMT markers such as ZEB1 and vimentin [72].

### 5.3. Irinotecan-Associated Resistance

Studies in colon-sphere cultures have demonstrated that miR-451 expression suppresses the ABCB1 pump responsible for the efflux of the chemotherapeutic agent irinotecan. In contrast, low levels of miR-451 were observed in patients who did not respond to irinotecan [73]. In irinotecan-resistant cell lines, reduced expression of miR-3664-3p has been associated with the increased expression of ABCG2 [74].

One of the mechanisms of drug resistance is the development of EMT, where overexpression of miRNA-376a-3p has been found to reprogram the EMT by reducing markers through IGF1R-induced cell survival and the PI3K/AKT pathway [75].

Table 1 and Figure 1 summarize the main miRNAs associated with chemoresistance in CRC and the underlying mechanisms.

## 6. microRNAs Related to Chemoresistance in a Different Clinical Stages

The main studies investigating miRNAs as potential markers for chemotherapy and treatment response in a clinical setting with patients are summarized below. The information is presented in Table 2.

### 6.1. MicroRNAs Related to Chemoresistance in Stage II and III Colorectal Cancer

In stages II and III, miRNA-based biomarkers may improve treatment outcomes by guiding adjuvant chemotherapy and radiotherapy decisions. miR-21 is strongly associated with 5-fluorouracil (5-FU) resistance in CRC through repression of mismatch repair (MMR) proteins, impairing DNA repair [76]. Its elevated expression correlates with poor neoadjuvant chemoradiotherapy response in rectal cancer [77]. A meta-analysis combining CRC cases across different stages has demonstrated that low expression of miR-143 is associated with higher event-free survival, while low expression of miR-145 is linked to poorer overall survival [78]. In CRC patients with stage II/III, miR-34a enhances radiosensitivity by triggering the cell cycle and cell apoptosis [79]. Despite previous studies revealing a promising role of miRNAs in chemoresistance during early stages (II or III), further research is needed to replicate a real clinical environment and compare their performance with more validated biomarkers such as MSI or ctDNA.

### 6.2. MicroRNAs Related to Chemoresistance in Stage IV Colorectal Cancer

Attempts have been made to identify miRNAs that are potential markers of chemoresistance in plasma or blood. However, the strategies used have not differentiated patients at various disease stages. This is a problem, given that CRC has very different treatment and prognosis in the early stages (non-metastatic or stages I to III) and depending on the location (colon or rectum) [4,5]. Nevertheless, some studies have assessed this specific group of patients, identifying certain miRNA candidates as potential chemoresistance markers. For example, a study in patients with rectal cancer (stages II–III) found that miR-21 expression in tissue could predict partial or complete response to neoadjuvant treatment or radiotherapy [80]. Previous studies in cell cultures found that this miR was associated with resistance to 5-FU due to the decreased expression of mismatch repair proteins (MMRs) [76]. A study in patients with CRC found that in FFPE tissue, high miR-625-3p expression was associated with worse rates of response to chemotherapy but not with prognosis regarding PFS. In addition, in cell cultures, oxaliplatin-resistant cells showed high expression of miR-625-3p, indicating that miR-625-p is associated with greater resistance to chemotherapy [81]. On the other hand, the downregulation of miR-377-3p has been associated with a worse prognosis in advanced CRC stages III–IV because this miRNA plays a role in the inhibition of ZEB2 through the Wnt/β-catenin pathway, factors that, when activated, promote the EMT, which is a mechanism of resistance to chemotherapy [82]. A study in stage IV patients evaluated miRNAs in blood that were associated with a poor response to FOLFOX. Using miRNA PCR arrays, five differentially expressed miRs were found; subsequent validation in a cohort of 72 patients revealed miR-19a as a biomarker of resistance [83]. Boisen (2014) studied miRNAs in FFPE samples from stage IV CRC patients treated with XELOX with or without bevacizumab (anti-VEGF). An exploratory analysis with PCR arrays and subsequent validation with RT-PCR revealed that high expression of miR-644-3p and low expression of miR-455-5p improved OS in the XELOX + bevacizumab group. In contrast, for the group treated with XELOX alone, high expression of miR-196b-5p and miR-592 indicated better OS [84]. In a retrospective study of a phase II trial that had evaluated the use of irinotecan + cetuximab as a third line of therapy, high expression of miR-345 in blood was found to be associated with a lack of response, suggesting this molecule as a potential resistance marker for these therapies [85]. One of the main mechanisms described in resistance to oxaliplatin is NER pathway activation. A retrospective study of a clinical trial (TRIBE trial) found that an SNP in the miRNA-binding domain of the RPA2 protein (belonging to the NER pathway) was associated with a better response regarding PFS in the group receiving oxaliplatin-based chemotherapy [86]. Finally, a retrospective analysis of the CAIRO study (capecitabine as monotherapy in metastatic CRC) found that higher miR-143 levels were associated with worse PFS [87], this being contrary to findings in cellular models [67,88]. One explanation by the authors is that CRC behaves differently in the early vs. late stages. In summary, miRNAs that could explain chemoresistance or chemosensitivity to current therapies in stage IV CRC have been identified.

**Table 2 ijms-26-01168-t002:** MicroRNAs implicated in chemotherapy resistance mechanisms in colorectal cancer: evidence from patient studies across all clinical stages as well as stage IV or metastatic disease. 5-FU: 5-fluorouracil; ypTNM: post-neoadjuvant TNM classification; OS: overall survival; PFS: progression-free survival.

Name	Expression Status	Characteristics	Outcome	Reference
Studies in Colorectal Cancer Patients Irrespective of Clinical Stage				
miR-21	Upregulated	Stages II–III (rectal cancer)	Worse pathological response (ypTNM) post-chemoradiotherapy	[76]
miR-21	Upregulated	Stage II–III (rectal)	Worse recurrence-free survival	[77]
miR-143	Downregulated	Stage I–IV	Better event-free survival	[78]
miR-145	Downregulated	Stage I–IV	Worse overall survival	[78]
miR-34a	Downregulated	Stage II–III	Associated with recurrence rate	[79]
miR-625-3p	Upregulated	Stages II–IV	Worse objective response rate	[81]
miR-377-3p	Downregulated	Stages I–IV	Correlation with more advanced stages	[82]
Studies in Stage IV or Metastatic Colorectal Cancer Patients				
miR-644-3p	Upregulated	Stage IV	Increased survival with XELOX + bevacizumab combination	[84]
miR-345	Downregulated	Stage IV	Increased survival with XELOX + bevacizumab combination	[84]
miR-196b-5p	Upregulated	Stage IV	Increased OS in patients treated with XELOX	[84]
miR-592	Upregulated	Stage IV	Increased OS in patients treated with XELOX	[84]
miR-143	Upregulated	Stage IV	Worse PFS in patients treated with capecitabine	[87]

## 7. Conclusions

The role of miRNAs in CRC chemoresistance highlights their potential as both biomarkers and therapeutic targets in precision oncology. As pivotal regulators of gene expression, miRNAs modulate key mechanisms underlying resistance, such as drug transport, apoptosis, and the EMT. Molecular insights into these processes offer a promising avenue to stratify patients, optimize chemotherapy regimens, and enhance outcomes for advanced CRC patients. However, significant challenges persist in translating miRNA research into clinical practice. Methodological inconsistencies, such as variability in sample sources, analytical techniques, and patient cohorts, complicate the validation and standardization of miRNA biomarkers. Moreover, the dynamic expression of miRNAs, influenced by tumor microenvironmental factors and systemic therapies, limits their reliability as predictive tools. These hurdles underscore the need for rigorous, large-scale studies with standardized protocols to bridge the gap between laboratory findings and clinical application.

Despite these challenges, the integration of miRNA profiling into clinical workflows holds transformative potential. By identifying patients likely to develop chemoresistance, miRNA-based strategies could guide personalized treatment, minimize toxicity, and improve survival rates. Nonetheless, controversies regarding their practical utility remain. For instance, while some miRNAs demonstrate clear associations with chemoresistance, data are conflicting, often due to differences in study designs or patient populations. This variability raises questions about the reproducibility and generalizability of findings across diverse clinical settings.

While miRNAs represent a promising frontier in CRC management, their clinical translation requires concerted efforts to address current limitations. Establishing robust validation frameworks and leveraging advancements in genomic technologies will be pivotal in unlocking their full potential as precision medicine tools. Ultimately, interdisciplinary collaboration will be warranted to refine methodologies and integrate miRNA-based insights into routine oncological care.

## Figures and Tables

**Figure 1 ijms-26-01168-f001:**
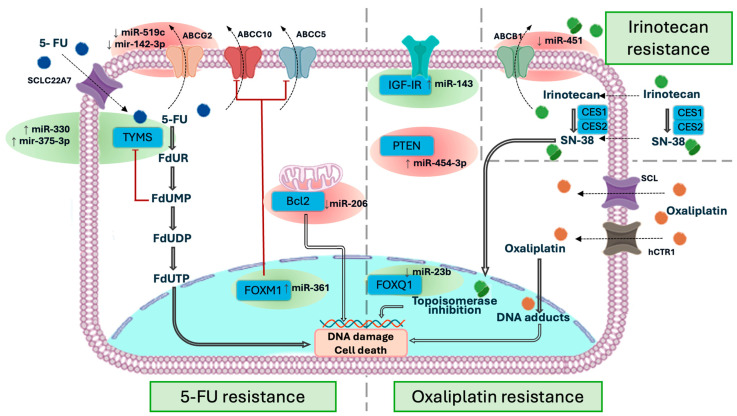
miRNA-associated mechanisms of resistance to 5-fluorouracil, oxaliplatin, and irinotecan. Each circle represents a key regulatory site modulated by a microRNA that promotes either chemosensitivity (green circles) or chemoresistance (red circles).

**Table 1 ijms-26-01168-t001:** MicroRNAs implicated in chemotherapy resistance mechanisms in colorectal cancer: evidence from preclinical and basic research models. 5-FU: 5-fluorouracil; OS: overall survival; PFS: progression-free survival.

Name	Expression Status	Drug	Effect	Mechanisms or Pathways Involved	Reference
miR-519c	Downregulated	5-FU	Chemoresistance	Increased ABCG2 expression	[60]
miR-142-3p	Downregulated	5-FU	Chemoresistance	Increased ABCG2 expression	[61]
miR-361	Upregulated	5-FU	Chemosensitivity	Inhibition of FOXM1 expression	[62]
miR-330	Upregulated	5-FU	Chemosensitivity	Inhibition of TYMS expression	[63]
miR-375-3p	Upregulated	5-FU	Chemosensitivity	Inhibition of TYMS expression	[64]
miR-27a	Upregulated	5-FU	Chemosensitivity	In vitro: inhibition of DPYD expression	[65]
miR-27a	Upregulated	Oxaliplatin	Chemoresistance	In silico: modulation of NER pathways	[66]
miR-206	Downregulated	5-FU	Chemoresistance	Increased Bcl-2 activity	[67]
miR-143	Upregulated	Oxaliplatin	Chemosensitivity	Inhibition of IGF-IR expression	[68]
miR-106a	Upregulated	Oxaliplatin	Chemosensitivity	Inhibition of FOXQ1 via TGF-β1 and Wnt pathways	[69]
miR-454-3p	Upregulated	Oxaliplatin	Chemoresistance	Inhibition of PTEN expression	[71]
miR-23b	Downregulated	Oxaliplatin	Chemosensitivity	Inhibition of EMT pathways (SNAI2 and vimentin)	[72]
miR-451	Downregulated	Irinotecan	Chemoresistance	Inhibition of ABCB1 expression	[73]
miR-3664-3p	Downregulated	Irinotecan	Chemoresistance	Inhibition of ABCG2	[74]
miRNA-376a-3p	Upregulated	Irinotecan	Chemosensitivity	Inhibition of IGF1R-induced cell survival, PI3K/AKT pathway	[75]

## Data Availability

Not applicable.

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
