# Peer review of "Understanding microRNA-Mediated Chemoresistance in Colorectal Cancer Treatment"

_ijms, 2025, doi:10.3390/ijms26031168_

Round 1
Reviewer 1 Report
Comments and Suggestions for Authors
This is a very well-written, cutting-edge, and comprehensive review of how miRNAs as biomarkers or therapeutic targets can potentially change the clinical course and prognosis of CRC.
Given that ijms-3432801 is a review article, there is no methodology or conclusion. Its main purpose is to summarize and analyze the current state of knowledge on the topic within a specific field.
The reference needs to be updated. Typically, references for a review should be within the last 3 years. Obviously, most of references in the manuscript are outdated. If the revision would switch to latest reference without modifying the content/body, I would be curious as to the author's reasons for using the old references in the first version.
I would recommend using the reference within 3-5 years.
If the authors would update the references, they will almost rephrase more than half of the manuscript.
Author Response
1 Reviewer #1: This is a very well-written, cutting-edge, and comprehensive review of how miRNAs as biomarkers or therapeutic targets can potentially change the clinical course and prognosis of CRC. Given that ijms-3432801 is a review article, there is no methodology or conclusion. Its main purpose is to summarize and analyze the current state of knowledge on the topic within a specific field. The reference needs to be updated. Typically, references for a review should be within the last 3 years. Obviously, most of references in the manuscript are outdated. If the revision would switch to latest reference without modifying the content/body, I would be curious as to the author's reasons for using the old references in the first version. I would recommend using the reference within 3-5 years. If the authors would update the references, they will almost rephrase more than half of the manuscript.
Response: We sincerely appreciate your thoughtful feedback. Many of the references in the initial version are more than five years old. We chose to include these earlier studies because they are pivotal in establishing the foundations of chemotherapy for colorectal cancer, as well as the concepts of chemoresistance and the use of miRNAs as biomarkers. In response to your recommendation, we have updated a significant portion of the references to more recent publications (i.e., published within the last five years), which required only minor revisions to the manuscript’s content. We have, however, maintained a select number of older references, as they remain fundamental to the understanding of chemoresistance mechanisms and miRNA biology in CRC.

Reviewer 2 Report
Comments and Suggestions for Authors
This manuscript focuses on the chemoresistance phenomenon of colorectal cancer (CRC). The molecular mechanisms involved in mediating chemoresistance of CRC were summarized, especially during therapy with chemotherapeutic drugs such as 5-FU, capecitabine, oxaliplatin, and irinotecan. And the alteration of serum or cellular miRNAs exposure to chemotherapeutic drugs and their potentially regulating effects for chemoresistance was analyzed. This review is interesting; However, some major concerns still need to be addressed.
1. In part 6, microRNAs related to chemoresistance in stage IV colorectal cancer was emphasized. However, chemotherapy was also an important therapeutic strategy for CRC patients at stage II~III. How about the effect of miRNA for predicting chemoresistance of CRC patients at stage II~III?
2. More information should be described in 5.3 for irinotecan-associated resistance.
3. For figure 1, the expression level of miRNAs (upregulation or downregulation) mediating chemoresistance shown in this figure should be provided. The arrow was recommended to indicate upregulation or downregulation.
4. For abbreviations in main text, the first appearance of abbreviation should be written in the style of full name plus abbreviation, such as KRAS/NRAS and BRAF.
Comments on the Quality of English LanguageCan be improved
Author Response
2 Reviewer #2: This manuscript focuses on the chemoresistance phenomenon of colorectal cancer (CRC). The molecular mechanisms involved in mediating chemoresistance of CRC were summarized, especially during therapy with chemotherapeutic drugs such as 5-FU, capecitabine, oxaliplatin, and irinotecan. And the alteration of serum or cellular miRNAs exposure to chemotherapeutic drugs and their potentially regulating effects for chemoresistance was analyzed. This review is interesting; However, some major concerns still need to be addressed.
In part 6, microRNAs related to chemoresistance in stage IV colorectal cancer was emphasized. However, chemotherapy was also an important therapeutic strategy for CRC patients at stage II~III. How about the effect of miRNA for predicting chemoresistance of CRC patients at stage II~III?
Response: Thank you for highlighting the importance of discussing chemoresistance in stage II–III CRC patients. In response, we have added a new subsection, Section 6.2, addressing the predictive and mechanistic roles of miRNAs in chemoresistance specifically in stage II and III disease. This section is supplemented by five new references, all of which are highlighted in yellow in the revised manuscript.
3 Reviewer #2: More information should be described in 5.3 for irinotecan-associated resistance.
Response: We appreciate your comments regarding the need for more information on irinotecan resistance mechanisms in CRC. We have added a new paragraphin Section 5.3. We conducted a new search on the role of miRNAs in potential regulatory targets involved in irinotecan resistance mechanisms, and we identified two new references that have been added to the text and the corresponding table.
4 Reviewer #2: For figure 1, the expression level of miRNAs (upregulation or downregulation) mediating chemoresistance shown in this figure should be provided. The arrow was recommended to indicate upregulation or downregulation.
Response: In line with your suggestion, we have modified Figure 1 to clearly indicate miRNA expression levels and their associated roles in chemoresistance. Arrows have been added to denote up- or downregulation, ensuring the figure is more informative and visually intuitive.
5 Reviewer #2: For abbreviations in main text, the first appearance of abbreviation should be written in the style of full name plus abbreviation, such as KRAS/NRAS and BRAF.
Response: We appreciate the reviewer’s comments. We have now provided full names for all abbreviations (e.g., KRAS/NRAS, BRAF) when they first appear in the text, with these updates highlighted in yellow for your convenience.

Round 2
Reviewer 1 Report
Comments and Suggestions for Authors
Reviewer's queries have been well addressed by the authors. The manuscript is fit for publication in the journal after due approval from Editor's desk.
Reviewer 2 Report
Comments and Suggestions for Authors
All the comments had been well answered, it can be published now.